# Clinically Applicable Pathological Diagnosis System for Cell Clumps in Endometrial Cancer Screening via Deep Convolutional Neural Networks

**DOI:** 10.3390/cancers14174109

**Published:** 2022-08-25

**Authors:** Qing Li, Ruijie Wang, Zhonglin Xie, Lanbo Zhao, Yiran Wang, Chao Sun, Lu Han, Yu Liu, Huilian Hou, Chen Liu, Guanjun Zhang, Guizhi Shi, Dexing Zhong, Qiling Li

**Affiliations:** 1Department of Obstetrics and Gynecology, The First Affiliated Hospital of Xi’an Jiaotong University, Xi’an 710061, China; 2Department of Obstetrics and Gynecology, Northwest Women’s and Children’s Hospital, Xi’an 710061, China; 3School of Automation Science and Engineering, Xi’an Jiaotong University, Xi’an 710049, China; 4Department of Pathology, The First Affiliated Hospital of Xi’an Jiaotong University, Xi’an 710061, China; 5Laboratory Animal Center, Institute of Biophysics, Chinese Academy of Sciences, University of Chinese Academy of Sciences, Beijing 100101, China; 6State Key Laboratory for Novel Software Technology, Nanjing University, Nanjing 210093, China; 7Pazhou Lab, Guangzhou 510335, China

**Keywords:** endometrial cancer, deep learning, screening, pathological diagnosis system, cell clumps

## Abstract

**Simple Summary:**

The soaring demand for endometrial cancer screening has exposed a huge shortage of cytopathologists worldwide. Deep learning algorithms, based on convolutional neural networks, have been successfully applied to the classification and segmentation of medical images. The aim was to establish an artificial intelligence system that automatically recognizes and diagnoses pathological images of endometrial cell clumps (ECCs). Total 39,000 ECCs (26,880 for training, 11,520 for testing and 600 malignant for verification) patches were obtained by the segmentation network. The training set reached 100% accuracy, the testing set gained 93.5% accuracy, 92.2% specificity, and 92.0% sensitivity. Therefore, an artificial intelligence system was successfully built to classify malignant and benign ECCs for reducing pathologists’ workload, providing decision-making assistance and promoting the development of endometrial cancer screening.

**Abstract:**

Objectives: The soaring demand for endometrial cancer screening has exposed a huge shortage of cytopathologists worldwide. To address this problem, our study set out to establish an artificial intelligence system that automatically recognizes and diagnoses pathological images of endometrial cell clumps (ECCs). Methods: We used Li Brush to acquire endometrial cells from patients. Liquid-based cytology technology was used to provide slides. The slides were scanned and divided into malignant and benign groups. We proposed two (a U-net segmentation and a DenseNet classification) networks to identify images. Another four classification networks were used for comparison tests. Results: A total of 113 (42 malignant and 71 benign) endometrial samples were collected, and a dataset containing 15,913 images was constructed. A total of 39,000 ECCs patches were obtained by the segmentation network. Then, 26,880 and 11,520 patches were used for training and testing, respectively. On the premise that the training set reached 100%, the testing set gained 93.5% accuracy, 92.2% specificity, and 92.0% sensitivity. The remaining 600 malignant patches were used for verification. Conclusions: An artificial intelligence system was successfully built to classify malignant and benign ECCs.

## 1. Introduction

Endometrial cancer (EC) has become the second most common malignant tumor in the female reproductive system, with about 378,400 new cases in 2018 worldwide [1]. With increasing life expectancy and altered living habits, the incidence of EC is on the rise, and patients tend to be younger [2,3]. The 5-year survival rate with appropriate treatment is more than 85% for localized, 49% to 71% for regional, and less than 17% for distant stages of EC [4]. Women exposed to high risks have been recommended to be screened. Screening for EC and precancerous changes has been strongly suggested for early diagnosis and to reduce morbidity and mortality [5].

Researchers on the early detection of EC focus on minimally invasive histopathologic and cytopathologic procedures [6]. An endometrial cytologic test (ECT) has been carried out in many countries, including Italy, the United States, and Japan. ECT was added into Japanese Law on health care for the elderly in 1987. The mortality from EC among Japanese high-risk women fell from 20% in 1950 to 8% in 1999 [7]. In the past 20 years, academics from different regions have put forward the invention and improvement of endometrial samplers and have recommended diagnosis systems for endometrial cytopathology [8,9,10]. Confirmed by diagnostic curettage, the sensitivity, specificity, and coincidence rate of a well-designed endometrial sampling device, Li Brush, were 92.73%, 98.15%, and 92.73%, respectively [11]. On the other hand, a large number of endometrial cytopathological slides need to be identified, which exposes the lack of pathologists.

With the development of artificial intelligence (AI) technology and the improvement of hardware computing power in recent years, deep learning (DL) in medical analysis is considered as a third eye for doctors [12]. DL algorithms, based on deep convolutional neural networks (CNNs), have been proven to strongly boost the development of biomedical image analysis [13,14]. CNNs are becoming a reference tool for pathologists and have been successfully applied to the classification and segmentation of medical images, reducing the workload of pathologists and providing decision-making assistance [15,16,17].

AI has been successfully used in recognizing pathologic images and identifying malignant and benign tumors. However, there are relatively few studies on EC recognition. In one study, a computer-aided morphology program was established to distinguish benign and malignant cells. Geometric and densitometric nuclear features were measured for analysis. However, the typical three-dimensional shape (crowded and overlapping nuclei) of the endometrium increased miscalculation [18]. In another experiment, an endometrial histopathological AI recognition system was built, though it had a relatively high false-negative rate because a few subtle features were undetectable at the cellular level [19]. Inspired by these studies, we developed a recognition system based on CNNs to automatically identify benign and malignant endometrial cell clumps (ECCs). The shortcomings of the two above studies will be overcome by analyzing the cellular clump’s structure and cytological characteristics.

## 2. Materials and Methods

### 2.1. Ethics Statement and Patients

The patients, who underwent curettage or hysterectomy, were recruited in the First Affiliated Hospital of Xi’an Jiaotong University from July 2015 to July 2020. This study was approved by the Ethics Committee of the First Affiliated Hospital of Xi’an Jiaotong University (XJTU1AHCR2014-007), and all patients signed written informed consent. The protocols were in compliance with the ethical principles for research that involves human subjects of the Helsinki Declaration for medical research [20].

Patients were excluded who had been diagnosed with suspected pregnancy or pregnancy, acute inflammation of the reproductive system, cervical cancer, or dysfunctional clotting diseases. Women with body temperature at or more than 37.5 °C were also excluded after being measured twice a day.

### 2.2. Preparation of Pathological Slides

We chose Li Brush (20152660054, Xi’an Meijiajia Medical Technology Co., Ltd., China) for endometrial cytological sampling (Figure 1a). Liquid-based cytology combined with Hematoxylin and Eosin staining was used for pathological slides of endometrial cells. The sampling, pathological slide, and staining procedures were described by Lu Han et al. [11]. Based on the endometrial cytological diagnostic criteria proposed by Chinese Expert Consensus [21], two experienced pathological professors (H.H. and G.S., with over 20 years of endometrial cytopathology experience) labeled all cytopathological slides and divided them into two classes: malignant (atypical cells of undetermined significance, suspected malignant tumor cells, and malignant tumor cells), and benign (non-malignant tumor cells). Slides with fewer than 10 or 5 ECCs were judged to be “unsatisfactory for evaluation” for premenopausal or postmenopausal women, respectively. Only a few isolated atypical or cancerous cells present were considered as satisfactory [22]. Histopathological diagnosis, acquired from the endometrium by curettage or hysterectomy, was regarded as the gold standard. Normal endometrium and endometrial hyperplasia without atypia were considered as benign; endometrial atypical hyperplasia and endometrial cancer were malignant. Only when consistent classification was reached between histology and the two pathologists’ cytology on a sample was the sample considered for the study. Otherwise, it was suspended [22].

### 2.3. Cytopathological Image Acquisition

We used a MOTIC digital biopsy scanner (EasyScan 60, 20192220065, Motic, Xiamen, China) to scan cytopathological slides (Figure 1b), using a lens with 200 times magnification (20×) to obtain whole slide images. A counterclockwise spiral scan was performed with a camera exposure time of 0.65 s per slide and automatic focal adjustment. Each scanned slide image was segmented into 1360 small images (1816 × 1519 pixels) (Figure 1b).

### 2.4. ECCs Image Annotation

Adobe Photoshop CC (2019 v20.0.2.30, Adobe Inc., San Jose, CA, USA) was engaged to sketch the edge of the ECCs. There is no doubt that ECCs from negative slides were all negative, but some ECCs were negative in positive slides. Thus, the two pathologists voted on the labels of each ECC again; when discordant voting results happened, they would have a discussion. If the discussion failed to conclude with an accurate diagnosis, the ECC was discarded. A benign diagnosis was defined as cell clumps with neat edges, nuclei with oval or spindle shape, and evenly distributed, finely granular chromatin [23,24]. Malignant diagnosis referred to a three-dimensional appearance, irregular (including dilated, branched, protruding, and papillotubular) edge, with the nucleus poloidal disordering or disappearing (including megakaryocyte appearance, nuclear membrane thickness, and coarse granular or coarse block chromatin) [23,25].

### 2.5. Segmentation Networks

The U-Net with jumping connection structure was selected to eliminate the interference of neutrophils and single cells, facilitating ECC extraction from each image. Figure 2 shows the U-Net architecture based on full convolutional networks. The U-Net architecture combined a down-sampling path to capture context and an up-sampling path to achieve precise localization. We calculated the probability that each pixel belonged to the cell clumps and normalized it. The collection of a detected cell clumps image was automatically marked as a region of interest (ROI) area. A total of 1000 images and their corresponding masks marked by pathologists were randomly selected for training. In order to describe the effect of the U-Net, we selected the Dice coefficient (a verification index of image segmentation accuracy) for evaluation.
$$\mathrm{Dice}=\frac{2|A\;\cap B|}{A+B}$$

The Dice coefficient is at the pixel level; A represents the area where the real target appears, and B signifies the target area that showed the predicted result (Figure 3a).

The segmented mask often has small holes and residues (Figure 3b). We used morphological operations (first corrosion and then expansion) to eliminate small holes. The ROI set was input into a subsequent neural network for endometrial cytopathological screening.

### 2.6. Data Preprocessing

We input the cytopathologic images into a trained U-Net to obtain the patch set of cell clumps. The segmentation results were first obtained by the U-Net, and background images (free single cells and white cells) were removed. Then, we extracted all the cell clumps using the minimum outer rectangle. The size of all cell clusters was uniformly resized to 256 × 256 by filling the surrounding area with pixels of value 0.

### 2.7. Classification Network

The CNNs were used to capture the characteristics of ECCs: nuclear heterogeneity, nuclear size, ratio between nucleus and plasma, chromatin homogeneity, cell polarity, isolation and aggregation of cell clumps, regularity of cell clump’s edge, etc. We constructed a DL model with DenseNet201 being the backbone to classify malignant and benign cell communities. The training set was annotated by two cytopathologists. The final fully connected layer of DenseNet201 was replaced by a global average pooling layer, then a single fully connected layer. The specific architecture is shown in Figure 4, and the output results were classified into two categories (Figure 1c). Then, the classification network was pre-trained on ImageNet. Several groups were carried out for comparative experiments to find the best patch input size and iteration time. The iteration was set to be 50, 100, 150, and 500 epochs in the training process. The results showed that the network converged at 100 epochs, and a longer training time was not necessary (Figure 5a). We changed the size of the input patch to 32 × 32, 64 × 64, 128 × 128, and 256 × 256, respectively (Figure 5b). When the input patch size was 256 × 256, the best result was achieved.

### 2.8. Network Evaluation

We conducted comparative experiments on four CNNs (VGG16, InceptionV3, ResNet, and DenseNet) and one Support Vector Machine (SVM). The hyperparameters, all kept consistent, were as follows: Loss function (Binary Cross-Entropy), Initial learning rate (0.0001), Learning rate delay (0.5), Batch-size (8), and Adam optimizer. In addition, the SVM classifier used a radial basic function kernel with parameters of 0.0078 and 2. DenseNet gained the best result due to its advantage of featured graph jump connection (Figure 5c–e).

All experiments were performed on a personal computer equipped with a GeForce GTX2080 super (NVIDIA) graphics processing unit. Python programming language 3.6.12 (Python Software Foundation, Wilmington, DE, USA) with keras 2.4.3 (Google Brain, Mountain View, CA, USA) and Tensor Flow 2.2.0 (Google Brain, Mountain View, CA, USA) for neural networks was used for the training.

### 2.9. Statistical Analysis

The following indexes were calculated by the four-lattice paired hypothesis test for statistical analysis: accuracy (Acc), sensitivity (Se), and specificity (Sp). The confusion matrix and receiver operating characteristic (ROC) curve were used to visualize the classification effect. The definition criteria were as follows:$$\mathrm{Acc}=\frac{\;\left(\mathrm{TP}\;+\;\mathrm{TN}\right)}{\;\left(\mathrm{TP}\;+\;\mathrm{FP}\;+\;\mathrm{TN}\;+\;\mathrm{FN}\right)}$$
$$\mathrm{Se}=\frac{\mathrm{TP}}{\;\left(\mathrm{TP}\;+\;\mathrm{FN}\right)}$$
$$\mathrm{Sp}=\frac{\mathrm{TN}}{\;\left(\mathrm{TN}\;+\;\mathrm{FP}\right)}$$

### 2.10. Plots and Charts

All the drawings were performed using the matplotlib package in Python and Matlab. The ROC curve of model performance was shown with specificity being the X axis and sensitivity being the Y axis. We used a bar chart to show the predictions from different CNNs and SVM. Line graphs were drawn to illustrate the results and compare performance between different groups.

## 3. Results

### 3.1. Baseline Characteristics

A total of 113 patients who met the criteria were enrolled for final analysis, among which 42 were malignant and 71 were benign. Table 1 lists the demographic data of these patients.

### 3.2. Dataset

A total of 15,913 annotated cell clump images were segmented on ×20 magnification digital slides. The average image size was 1816 × 1519 pixels by width and height. We used a trained U-Net to extract ECC patches from the 15,913 images and obtained 39,000 ECC patches. Divided in 7:3, 26,880 and 11,520 patches were used for training and testing. The remaining 300 benign patches and 300 malignant patches were included in a verification set.

### 3.3. Verification Set and Test Set

The prediction results of ECC patches were completely in accordance with the labels given by the pathologists. We randomly exhibit the results of eight (three malignant and five benign) validation patches (Figure 6A).

In the test set, the accuracy and specificity of the classifier were 93.5% and 92.2%, respectively. The DenseNet achieved a 95.1% area under the curve score (AUC). In addition, we compared the results with four other common classification models (Figure 5c–e).

### 3.4. False Results

DenseNet obtained a 5% false-positive rate and an 8% false-negative rate in the test set (Figure 5c). We randomly listed eight common failure patches in the test set. The six false-negative (missed diagnosis) patches included one well-differentiated endometrial adenocarcinoma, three endometrial atypical hyperplasia, and two poorly differentiated endometrial adenocarcinomas. In addition, two over-diagnoses occurred (Figure 6B).

### 3.5. Data Supporting

The results of this study are available from the corresponding authors (Qiling Li and Dexing Zhong). Because of hospital policy, the data cannot be made public.

## 4. Discussion

### Principal Findings

For the first time, we introduced two neural networks based on deep convolution, namely U-Net and DenseNet, to segment ECC images and recognize patches, respectively. The DenseNet achieved 93.5% accuracy and 92.2% specificity. At the same time, this system was developed for screening, and the sensitivity of our algorithm was better than that of all the comparison ones, reaching 92.0%. The results indicated that the neural network has great feasibility and potentiality in endometrial pathological image recognition.

## 5. Results

It is well-known that a large amount of labeled data is often required to train a high-quality machine learning classifier through DL to complete a specific cancer classification task [26]. Due to the high amount of time and effort required for image annotation work, as well as the protection of patients’ privacy, there are currently few endometrial image datasets available to the public. Despite the limited dataset, our classifier performed well in the 10-fold cross-validation and in the external validation of 15,913 images. 

### 5.1. Clinical Implications

At the beginning of the experiment, we considered that DL was able to automatically learn cancer’s information from pathological images [27]. We put the unlabeled benign and malignant images into the network for recognition and obtained 40–70% specificity (data not shown) from multiple networks, proving the method to be a failure. ECCs are quite different from non-cellular clumps in ecological appearance, cell morphological structure, and other pathological characteristics. U-net combines low-resolution information (to provide the basis for object category recognition) and high-resolution information (to provide the basis for precise segmentation and positioning), which is perfectly suitable for medical image segmentation. Combined with the pathological features in patients with ECCs, we chose the U-Net as the segmentation network to analyze and calculate the probability that each pixel belonged to the cell clumps. The detected cell clump images were automatically marked as ROI areas. The obtained ROI set was processed by a traditional image-processing algorithm to eliminate small holes. The ROI set was input into a subsequent neural network for cytopathological screening of the endometrium. We built a DL model with DenseNet201 as the backbone. The DL model was trained by the dataset annotated by cytopathologists, and the model was built to classify malignant and benign cell clumps. It turned out that our model alleviated the vanishing gradient problem, strengthened feature propagation, encouraged feature reuse, and outperformed ResNet50 with the same number of parameters. In order to compare the prediction performance of various DL algorithms on an experimental dataset, four commonly used CNNs were used to train different classifiers, namely VGG16, Inception-v3, ResNet, and DenseNet. The SVM classifier, which used features extracted by the CNN as input, had a better performance than the end-to-end CNN classifier [28,29]. Therefore, on the basis of previous experiments, the DenseNet had the best performance in extracting sample features to train the SVM classifier. In addition, a group of comparative experiments were also performed with traditional PCA + SVM machine learning method.

The results of the test set showed that the false-negative rate was twice as high as the false-positive rate. We analyzed all the missed and over-diagnosis images and randomly selected eight patches to illustrate the common error that occurred. There were two false positives: one patch of secretory phase endometrium and one patch of complex hyperplasia. One reason for this was that the endometrial cells were clustered and seriously overlapped. It was difficult to distinguish well-differentiated EC from the proliferative endometrium, and it was difficult to distinguish complex hyperplasia from atypical hyperplasia. Another reason was that the dysplasia coincidence rate between the cytological and histological pathological diagnosis was relatively low, which was 56% in some studies [30]. This was the main reason for their miscalculation.

### 5.2. Research Implications

Due to the development of liquid-based cytology and endometrial cell sampling in recent years, ECT has been gradually accepted as a simple, rapid, and economical endometrial screening method [31]. Moreover, AI can be applied to the pathological recognition of endometrial cells to promote screening. AI works steadily and indefatigably, and can quickly screen out suspicious malignant results, allowing pathologists to focus on the malignant results and improve the accuracy and efficiency of diagnosis [32].

### 5.3. Strengths and Limitations

This study had some limitations. First, although our images were labeled in a randomized and blind way, and histological diagnosis was used as control, and the two pathologists’ diagnoses were still somehow subjective. We hope that more recommendations from pathologists in different treatment centers will be included in follow-up studies regarding the proposed diagnostic system. Second, liquid endometrial cytological smear was used in our diagnostic system. At present, cell block technology can prepare slides with cell clumps and micro tissues, which is expected to further refine the diagnostic results and provide better diagnosis and treatment suggestions for clinical work [33]. We will focus on improving the performance of the classifier by training it with more samples, aiming at subdividing endometrial pathological types in future research.

## 6. Conclusions

This study confirmed that the recognition of DL has similar specificity and sensitivity to manual diagnosis. At the same time, the DL saves time and manpower. Therefore, the use of endometrial liquid-based cytology in combination with AI to identify ECC is reliable for EC screening and is able to reduce pathologists’ workload. By carrying out this form of screening work, cross-population, big data will be rapidly established, and the participation of scholars from different regions will greatly promote the development of precision medicine.

## Figures and Tables

**Figure 1 cancers-14-04109-f001:**
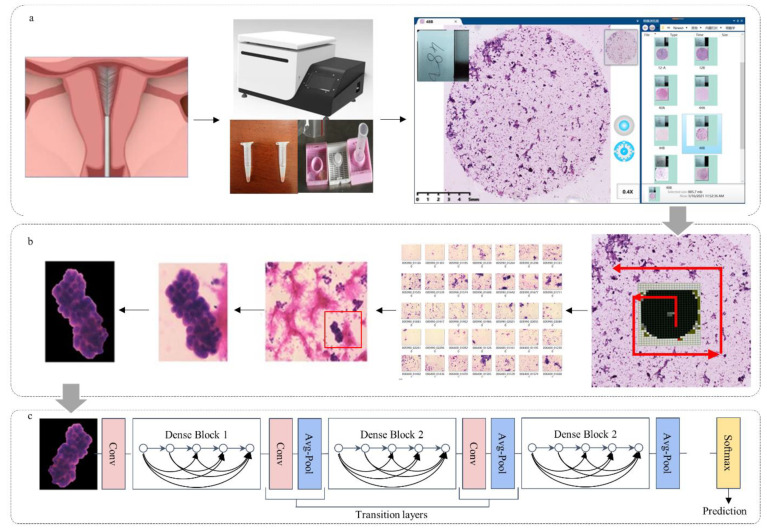
The process of obtaining images and recognition. (**a**) Sampling procedure; (**b**) cytological slides diagnosis; (**c**) classification using endometrial cytological images feature.

**Figure 2 cancers-14-04109-f002:**
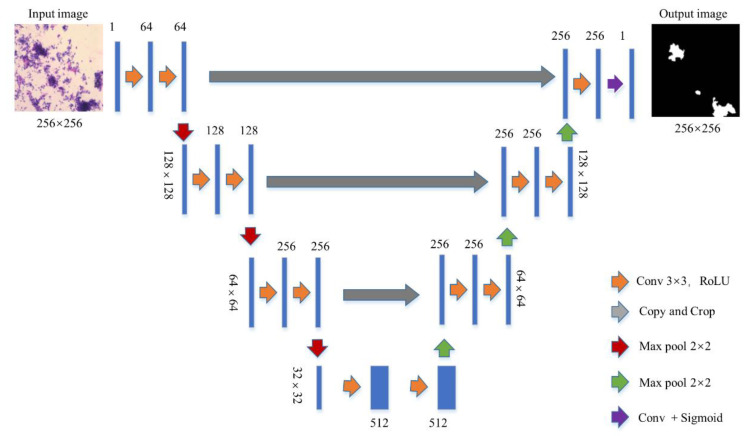
Segmentation network. The blue box represents the feature map. The yellow arrow represents 3 × 3 convolution and striding of 1 used for feature extraction; we set the padding as 1 to ensure that the size of the convolutional image at the same steps was stable. The gray arrow indicates skip-connection, which is used for feature fusion, and pure up-sampling will cause the loss of information. The red arrow indicates the 2 × 2 maximum pooling, which is used to reduce the dimensionality. The green arrow indicates up-sampling, which is used to restore the dimension. The cyan arrow indicates the convolution plus activation function, which is used to output the result.

**Figure 3 cancers-14-04109-f003:**
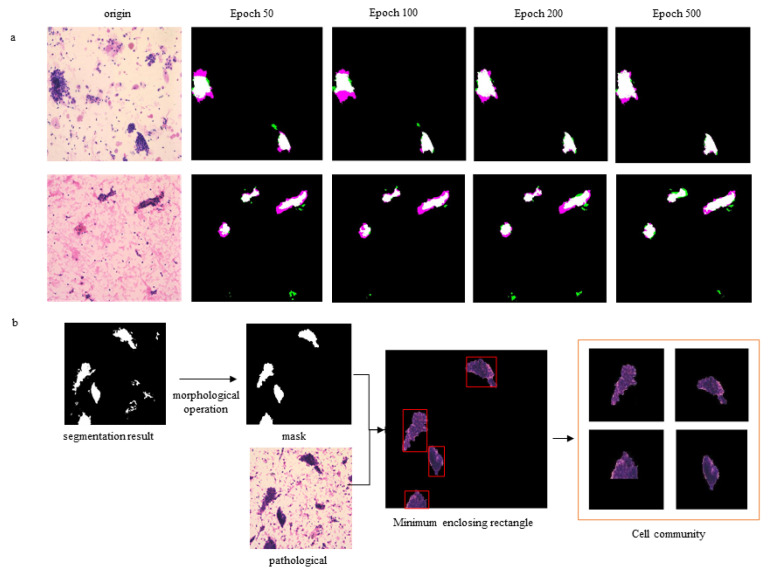
The effect of segmentation. (**a**) Variation of segmentation accuracy with training epochs. Compared with the ground truth (mask was manually marked by the physician), the red areas were not predicted in the mask of the model training; compared with the ground truth, the green areas represent other predicted areas in the mask of model training. (**b**) The process of ECC acquisition.

**Figure 4 cancers-14-04109-f004:**
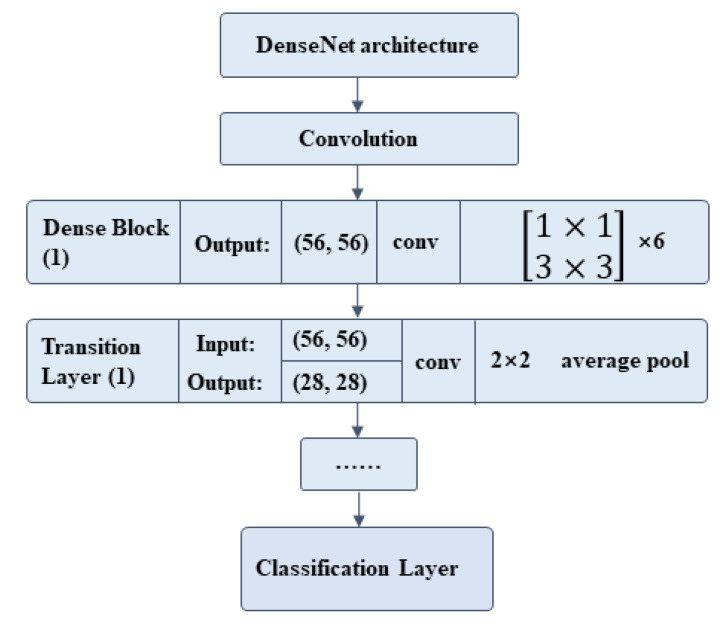
The recognition network architecture for classifying endometrial cell clusters. The size of the input image is 256 × 256, and each 3 × 3 convolution is preceded by a 1 × 1 convolution operation.

**Figure 5 cancers-14-04109-f005:**
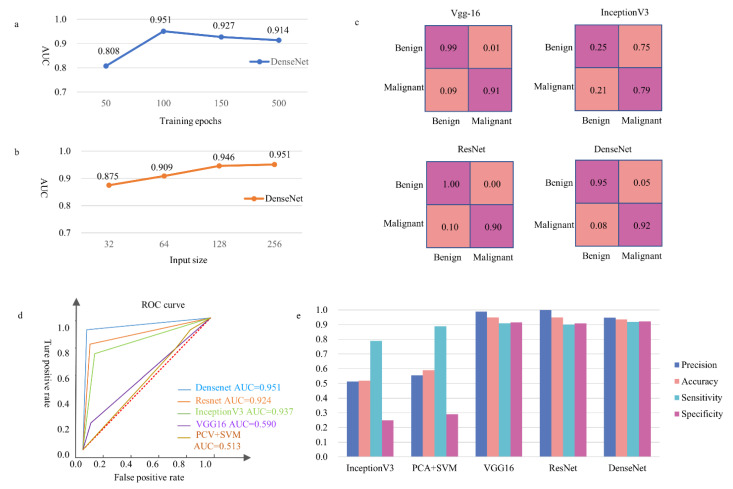
The performance of our model and four other common DL models on the same validation set. (**a**) Description of the AUC corresponding to the network with different numbers of iterations. (**b**) Description of the AUC corresponding to the network with different image input sizes. (**c**) The confusion matrix of different networks under the same hyperparameter conditions. The horizontal axis was a true label, the vertical axis was the predicted label, and the lower false-negative rate was preferred. (**d**) The ROC curves of different models. (**e**) The precision, accuracy, sensitivity, and specificity of different models.

**Figure 6 cancers-14-04109-f006:**
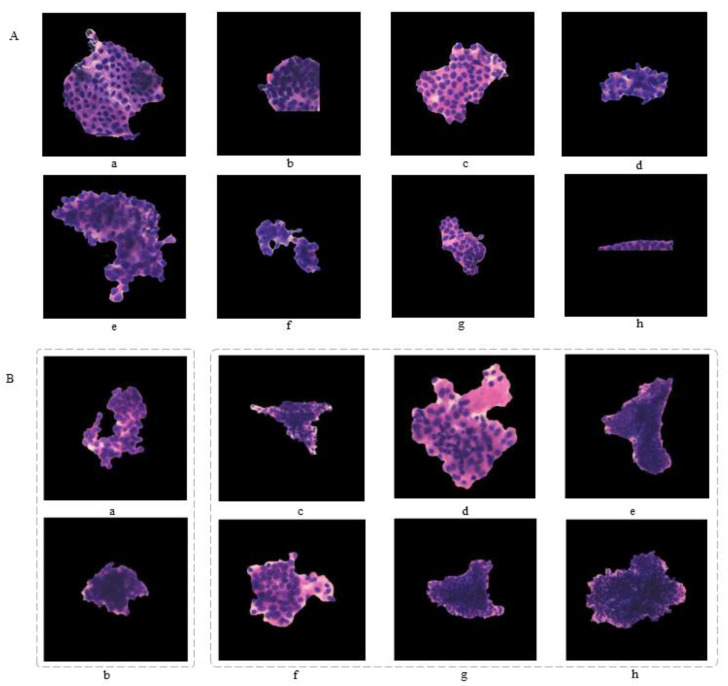
Presentation of true and false results. (**A**) A 100% consistency of results was achieved in the training set. Patches (**a**–**c**) showed the true positive, and patches (**d**–**h**) showed the true negative. (**B**) Analysis of false results in test set. The two false-positive (over diagnosis) patches (**a**,**b**) are exhibited. The six false-negative patches included one well-differentiated endometrial adenocarcinoma (**c**), three atypical hyperplasia (**d**–**f**), and two poorly differentiated adenocarcinomas (**g**,**h**).

**Table 1 cancers-14-04109-t001:** Patients characteristics.

Characteristics	n
**Source**	
Inpatient Department	66
Outpatient Department	47
**Age**	
<40 years old	13
≥40 years old	100
**Menstrual Status**	
Premenopausal	51
Postmenopausal	66
Abnormal uterine bleeding	35
**Other Disease**	
Ovarian cancer	0
Hypertension	10
Diabetes	4
Hormone replacement therapy	1

## Data Availability

The raw data supporting the conclusions of this manuscript will be made available by the authors without undue reservation to any qualified researcher. All data generated or analyzed during this study are included either in this article or available from the correspondence authors.

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
