# Peer review of "Clinically Applicable Pathological Diagnosis System for Cell Clumps in Endometrial Cancer Screening via Deep Convolutional Neural Networks"

_cancers, 2022, doi:10.3390/cancers14174109_

Round 1

Reviewer 1 Report

I am satisfied with the revised version of manuscript and recommend this article for publication. 

Reviewer 2 Report

Dear Authors,

The Rebuttal against the queries are satisfactory and compregensive

This manuscript is a resubmission of an earlier submission. The following is a list of the peer review reports and author responses from that submission.

Round 1

Reviewer 1 Report

Dear Author,

Your manuscript is interesting to read.

1.In material and  method, section 2.4 the image annoation needs to be more detailed focused on how both the pathologists who annoated the images has been calibrated. what was intra-observer and interobserver agreement??

2. I am curious regarding section 2.3 , you described that slide is enlarged, it might have lost the pixel information which might have compromised the data quality?? Please explain.

3. Regarding section 2.5, is there any standard protocol you followed to classify the clump cells you mentioned.??

4. The manuscri[pt needs to be edited by native Englsih speaker for Englsih language and grammar.

Author Response

  1. In material and method, section 2.4 the image annoation needs to be more detailed focused on how both the pathologists who annoated the images has been calibrated. what was intra-observer and interobserver agreement??

Response: It’s a good advice and we added 5 more sentences in page 5 line 126-127: “The annotation procedure of the image comprised three steps, labeling stage, verification stage, and check stage. First, slides were randomly assigned to two pathologists for labeling. Second, to ensure the accuracy of the training set, we iterated training tests for difficult cases. In the final step, slides with inconsistent diagnoses would be sent back for reidentification.”

  1. I am curious regarding section 2.3, you described that slide is enlarged, it might have lost the pixel information which might have compromised the data quality?? Please explain.

Response: You must have read our manuscript very carefully. Thank you very much for your advice. We have made a small mistake here, which led to your misunderstanding, for which we apologize. In page 5 line 118-120,“We performed the MOTIC Digital Biopsy Scanner (EasyScan 60, 20192220065, China) to scan the cytopathological sections (Figure 1-b), and the slide was enlarged by 20 times”, the cytopathological sections were enlarged by 20 times to make pathological slides. Therefore, we changed it to “We performed the MOTIC Digital Biopsy Scanner (EasyScan 60, 20192220065, China) to scan the cytopathological sections (Figure 1-b), and it was enlarged by 20 times to make slides.”

  1. Regarding section 2.5, is there any standard protocol you followed to classify the clump cells you mentioned.??

Response: The standard protocol we followed was mentioned in section 2.4. The description of the cell clumps is presented separately in section 2.5. I apologize for the misunderstanding caused by paragraph arrangement. We moved the standard protocol from section 2.4 to section 2.5. “ECC images.

The pathologic diagnostic criteria were according to the International Society of Gynecological Pathologists, and were referenced Classification of uterine tumors by the World Health Organization in 2014(21). The cytopathologic and histopathologic diagnosis were divided into two categories: malignant group (EC tissue, EC cells, tissue of endometrial atypical hyperplasia, and endometrial atypical cells), and benign group. Slides with fewer than 10 cell clumps were judged to be “unsatisfactory for evaluation”. Only a few isolated atypical or cancerous cells present were considered as satisfactory (22). Benign diagnosis was defined as cell clumps with neat edges, nuclei with oval or spindle shape, and evenly distributed chromatin with finely granular (23, 24). Malignant diagnosis referred to a three-dimensional appearance, irregular (including dilated, branched, protruding and papillotubular) edge, with nucleus poloidal disordering or disappearing (including megakaryocyte appearance, nuclear membrane thickness, and coarse granular or coarse block chromatin)(23, 25).

  1. The manuscript needs to be edited by native English speaker for English language and grammar.

Response: Thank you for your valuable comments. We have asked a well-established expert to polish our manuscript to meet the English presentation standard.

Reviewer 2 Report

The manuscript entitled “Clinically applicable pathological diagnosis system for cell clumps of endometrial cancer screening via deep convolutional neural networks” by Qing Li et al., shared some interesting findings regarding the use of AI to screen the Endometrial cancer samples. Given the increasing trend in cases of EC worldwide, there is urgent need of improving the current screening system to reduce the female mortality rate.

I have some minor suggestions to include in current manuscript:

·       What is the opinion of authors to include cancer markers criteria?

·       What additional measures can be included to reduce the false negative rate.

Round 2

Reviewer 1 Report

Dear Author,

The rebuttals against the queries are unsatisfactory